# Awareness of Fluid Losses Does Not Impact Thirst during Exercise in the Heat: A Double-Blind, Cross-Over Study

**DOI:** 10.3390/nu13124357

**Published:** 2021-12-03

**Authors:** Catalina Capitán-Jiménez, Luis F. Aragón-Vargas

**Affiliations:** 1Human Movement Science Research Center, University of Costa Rica, Montes de Oca, San José 11-501-2060, Costa Rica; luis.aragon@ucr.ac.cr; 2Department of Nutrition, Hispanoamerican University, El Carmen, San José 10101, Costa Rica

**Keywords:** voluntary fluid intake, dehydration, thirst perception

## Abstract

Background: Thirst has been used as an indicator of dehydration; however, as a perception, we hypothesized that it could be affected by received information related to fluid losses. The purpose of this study was to identify whether awareness of water loss can impact thirst perception during exercise in the heat. Methods: Eleven males participated in two sessions in random order, receiving true or false information about their fluid losses every 30 min. Thirst perception (TP), actual dehydration, stomach fullness, and heat perception were measured every 30 min during intermittent exercise until dehydrated by ~4% body mass (BM). Post exercise, they ingested water ad libitum for 30 min. Results: Pre-exercise BM, TP, and hydration status were not different between sessions (*p* > 0.05). As dehydration progressed during exercise, TP increased significantly (*p* = 0.001), but it was the same for both sessions (*p* = 0.447). Post-exercise water ingestion was almost identical (*p* = 0.949) in the two sessions. Conclusion: In this study, thirst was a good indicator of fluid needs during exercise in the heat when no fluid was ingested, regardless of receiving true or false water loss information.

## 1. Introduction

Hydration is a factor to take into consideration for health and performance during prolonged exercise, especially in the heat, when fluid replacement of sweat losses is relevant. Thirst has been widely studied as a mechanism of hydration control during exercise, but whether it is good enough to maintain euhydration is still controversial [1,2,3,4].

Previous studies have shown that thirst perception is strongly associated with actual fluid deficits during exercise, provided no fluid ingestion is allowed [5,6]. Manipulation of thirst perception has been studied to see how it can affect performance [7,8], using protocols to control thirst through saline infusions and mouth rinsing. Other behaviors and perceptions may be modified by external variables, such as publicity, noise, and vibrations. The so-called Hawthorne effect [9], as an example, is normally understood to describe how the behavior of workers—or patients—is modified when they become aware of being observed. Thirst may be easily affected by environmental conditions, fatigue, and especially by drinking [2]. Studies have clearly shown that the major issue with thirst-driven intake is a rapid decrease of the desire to drink after fluid ingestion, even when people replace less than 60% of what they lose [5,10]. We do not know, however, whether thirst perception can be intentionally modified by providing information or whether humans will change their perceptions of thirst just because they know how much their fluid losses are. Thirst perception, and its corresponding behavior, drinking, may be susceptible to change depending on awareness of fluid losses.

Therefore, the aim of the study was to identify whether thirst perception (TP) can be affected by awareness of fluid losses incurred during exercise in the heat.

## 2. Materials and Methods

The current investigation used a double-blind, cross-over design to determine whether thirst perception (TP) can be affected by receiving information about fluid losses during exercise in the heat. Subjects completed two randomly assigned sessions. Experimental testing procedures required subjects to exercise in the heat until they dehydrated by ~4% BM. Subjects were asked to report using the thirst perception scale every 30 min from the onset of exercise until they reached the target level of dehydration, and then to drink as much as they wanted for 30 min.

### 2.1. Subjects

Eleven apparently healthy, physically active males participated in the study. The sample size of the study was determined from previous studies that were similar in design [10,11]. Informed consent was obtained from all subjects involved in the study. The protocol was approved by the Institutional Ethics Committee.

### 2.2. Procedures

In one session, subjects received real information (RI) about their fluid losses, and in the other session they received information corresponding to 60% of their real fluid losses (false information, FI), which is close to the average voluntary drinking reported in other studies [10,11]; sessions were randomly assigned. Each participant arrived in the laboratory after overnight fasting, performed the baseline procedures, exercised in the heat, and rehydrated ad libitum. At different points during the protocol, self-reported measures were obtained for thirst, fullness, and heat perception.

On testing days, participants reported to the laboratory and voided their bladders completely. Urine was collected and analyzed with a refractometer for urine-specific gravity (manual refractometer ATAGO^®^ model URC-Ne, Minato-ku, Tokyo, Japan, with a spectrum of 1.000 to 1.050). Urine osmolality (U_osm_) was also measured via freezing point depression (Advanced Instruments 3250 osmometer; Norwood, MA, USA). Nude baseline body weight was measured to the nearest 10 g (e-Accura^®^ scale, model DSB921, Qingpu, Shanghai, China).

Self-reported thirst was recorded with a visual analog scale. The scale consisted of a continuous 100 mm line with a mark on the left end indicating “not at all,” and on the right “extremely,” corresponding to the question, “How thirsty do you feel?” Perceived heat sensation was measured with an analog scale from “1: incredibly cold” to “8: incredibly hot.” Finally, for the feeling of fullness, the question was: “How full do you feel?” Scoring between 1 (not at all) and 5 (very, very) was used. This group of scales had been used in previous studies [10,11].

Baseline measurements were taken for both sessions upon arrival to the laboratory. These consisted in nude body weight, urine-specific gravity, urine osmolality, and perceptions of thirst, heat, and fullness. Participants were asked to use the same clothing for both sessions.

Each participant ingested a standardized breakfast after baseline measurements (750 kilocalories: 24.6% fat, 20.7% protein, and 54.7% carbohydrates; 250 mL of fluid; 1500 mg sodium). After resting for thirty minutes, baseline measurements were taken, and the exercise session started.

In both sessions, each participant exercised intermittently (30 min bicycle/30 min treadmill, at 70–80% HRmax) in the heat (WBGT = 28.8 ± 0.1 °C and 28.9 ± 0.3 °C, for RI and FI, respectively; T = 32.5 ± 0.7 and H = 73 ± 3 for RI, and T = 32.2 ± 1.1 and H = 70 ± 3 for FI), to a target dehydration equivalent to 4% body mass (BM). Subjects were weighed every 30 min to monitor their fluid losses; after every weighing, subjects received information according to the session. Thirst perception was measured every 15 min after they received information. Water ingestion during exercise was not allowed. Heat stress was monitored with a Questemp36^®^ monitor (3M, Oconomowoc, WI, USA).

To achieve the double-blind design of the study, an assistant was responsible for monitoring body weight and providing the information about weight losses to the participants; he did not know the objective of the study. This assistant measured body weights and passed the information on to the researchers outside the chamber, who performed the calculations of weight loss that had to be communicated to the participants. The participants were weighed naked behind a curtain; therefore, the scale display was not visible to them. This ensured that they could only obtain information from the assistant. Both the participant and the assistant in the chamber were informed of the real aim of the study upon completion.

Upon exercise termination, participants were instructed to drink as much as they needed from previously weighed bottles for 30 min. Water intake was measured with an OHAUS^®^ Compact Scales, model CS2000 (Parsippany, NJ, USA) food scale. Urine-specific gravity (USG) and osmolality (U_osm_), fullness, heat sensation, and thirst perception (TP) were measured pre- and post-exercise, and post-rehydration.

### 2.3. Statistical Analysis

Mean and standard deviation were used for descriptive statistics. Normality was checked for all variables. A t-test was performed to identify differences between sessions for each variable (body mass, USG, Uosm, thirst, WBGT, fullness, and heat sensation). One-way analyses of variance were performed to see differences over time for each variable (urine osmolality, thirst, heat sensation, and fullness). Where ANOVA showed a statistically significant main effect, Tukey’s post hoc tests were performed to compare time differences.

## 3. Results

Participants were 23.0 ± 3.0 years old, 1.75 ± 0.07 m tall, and weighed (upon arrival) 76.7 ± 4.9 kg. Pre-exercise conditions were the same for both sessions; see Table 1.

Participants exercised for 110.0 ± 24.8 or 115.0 ± 22.3 min (t = −1.27; *p* = 0.232) during the RI and FI sessions, respectively, and achieved body mass losses of 2.98 ± 0.37 kg and 2.93 ± 0.33 kg, respectively, representing actual dehydration of 3.88 ± 0.43% and 3.81 ± 0.38% (t = −0.30; *p* = 0.756), respectively. Subjects ingested the same amounts of water at the ends of the sessions (1220 ± 249 mL and 1228 ± 422 mL; t = −0.66, *p* = 0.949). At the end of the rehydration period, hypohydration was still equivalent to 2.50 ± 0.48% or 2.48 ± 0.68% of pre-exercise body mass, respectively.

Figure 1 shows urine osmolality between conditions over time, pre-exercise (RI: 654.3 ± 296.4 and FI: 663.2 ± 297.4), post-exercise (RI: 630.1 ± 295.5 and FI: 579.1 ± 279.3), and after rehydration (RI: 695.2 ± 259.5 and FI: 665.9 ± 288.5). Uosm was not different between sessions (f = 0.134; *p* = 0.722), and there was no difference over time (f = 0.65; *p* = 0.804) and no interaction (f = 0.243; *p* = 0.633) either.

A strong and significant association was found between the perception of thirst and the percentage of dehydration in both sessions (r = 0.992 and r = 0.979, *p* < 0.05, for RI and FI, respectively)

Thirst perception showed no difference between sessions (F = 0.661; *p* = 0.447). There was a difference over time (F = 44.6; *p* = 0.001) pre-exercise, but no interaction (F = 0.382; *p* = 0.559). Percentages of dehydration did not differ between sessions (t = −0.30; *p* = 0.756). Fullness showed no differences between sessions (F = 3.74; *p* = 0.205), nor over time (F = 3.74; *p* = 0.304). Meanwhile, sensation of heat did not differ between sessions (F = 0.982; *p* = 0.360) or over time (F = 2.88; *p* = 0.140). See Figure 2A–D.

## 4. Discussion

The aim of the study was to identify whether thirst perception (TP) can be affected by awareness of fluid losses during exercise in the heat. The main finding of this study was that thirst perception during exercise in the heat was not influenced by true or false information about fluid losses. After exercise, subjects drank one-third of their losses (≈1.2 L), a large volume for 30 min of rehydration, independently of the information they received. Thirst perception was markedly reduced at this point.

This study design differs from others because we manipulated thirst through the information of fluid losses of the subjects, contrary to others that manipulated thirst with saline infusions, mouth rinsing, or small quantities of water [7]. We also focused only on thirst perception during exercise and not on performance, which was done by other studies [5,12,13,14]. This could be relevant because an important proportion of the physically active population may be relying on thirst to drive their hydration, while not caring much about performance. Athletes, who care about performance, represent a small percentage of the population. This study confirms that thirst perception can detect dehydration and it will go higher as the level of dehydration increases during exercise in the heat (Figure 2A,B). However, as has been shown in other studies, this association between thirst perception and hydration needs quickly deteriorates as soon as subjects drink some fluid [5,6,10,12,15]. In our case, after 30 min of ad libitum rehydration, TP was markedly reduced, but the fluid deficit remained high. This behavior of thirst perception has been reported in previous studies [7,10,11]. Others have shown that thirst decreases rapidly as soon as liquid is ingested, long before the fluid lost during exercise has been replenished [2,7,8]. In the present study, thirst perception was evaluated at the end of the protocol, 30 min after finishing exercise. Unfortunately, the protocol used in the present study was focused on thirst responses during exercise, and post-exercise fluid intake and TP were only recorded once over those 30 min. We acknowledge that it would have been interesting to follow TP for a longer period, to complement the information on reduced ad libitum water intake reported by others [2,5,7].

Even when WBGT and exercise intensity were high, thirst perception between sessions was the same and had the same behavior over time, regardless of receiving true or false information about fluid loss, reflecting the percentage of dehydration [10] (see Figure 2A,B). It should be noticed that in this study, drinking during exercise was not allowed. We expect that this behavior will change when drinking or mouth rinsing is allowed, as others have shown [10,16]. Possibly, knowing that drinking would not be possible until the end of exercise could alter thirst perception, but this was a necessary element in the study design.

Thirst perception is widely used as a reference for hydration needs, especially in physically active persons (not necessarily athletes). Moreover, thirst could be used as a parameter as long as no liquid is ingested during exercise [7,10]. From this particular study, it may be added that internal signals such as increases in plasma osmolality or decreases in plasma volume [13,17] seem to be adequate to reflect dehydration in the absence of drinking, despite inaccurate external information that a person may receive about his hydration status. However, this does not mean that thirst perception should be recommended as a hydration strategy during exercise, due to its limitations whenever fluid is ingested [7,10]. As a hydration strategy during exercise, a pre-established protocol seems to be a better alternative.

## 5. Conclusions

In conclusion, thirst perception (TP) was not affected by receiving information about fluid losses during exercise in the heat in the absence of drinking. This might suggest that awareness of fluid losses during exercise cannot override the dehydration-induced hypothalamic signal for thirst.

## Figures and Tables

**Figure 1 nutrients-13-04357-f001:**
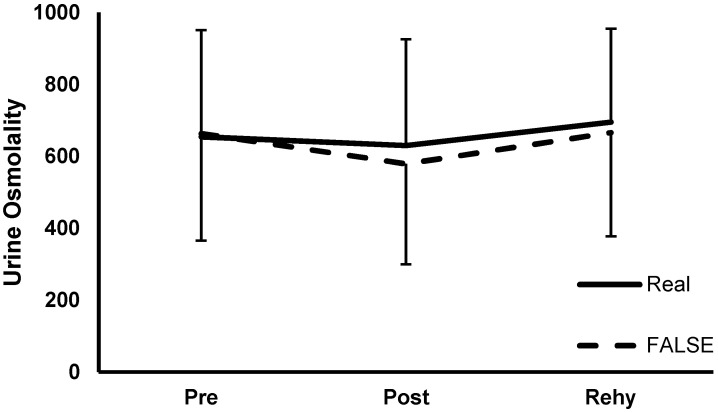
Urine osmolatity values (mean ± s.d): no difference between sessions (*p* = 0.722) or over time (*p* = 0.804). PRE = pre-exercise. POST = post-exercise. REHY = upon completion of rehydration.

**Figure 2 nutrients-13-04357-f002:**
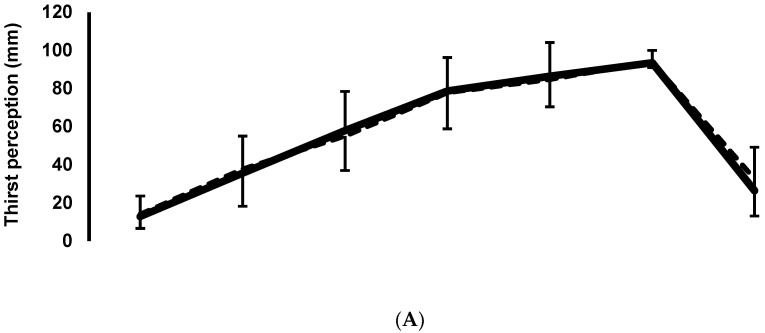
Values shown are mean ± s.d. (**A**) Thirst perception shows no difference between sessions (*p* = 0.447). There is a difference over time (*p* = 0.001) for pre-exercise, but no interaction (*p* = 0.559). (**B**) Percentage of dehydration did not differ between sessions (t = −0.30; *p* = 0.756). (**C**) Fullness: no differences between sessions (*p* = 0.205), nor over time (*p* = 0.304). (**D**) Heat sensation did not differ between sessions (*p* = 0.360) or over time (*p* = 0.140). By design, not all subjects finished at the same time: at 120 min, *n* = 9; at 150 min, *n* = 2; at REHY *n* = 11. Real: real information trial. False: false information trial, equivalent to 60% of actual water loss.

**Table 1 nutrients-13-04357-t001:** Pre-exercise conditions for each session.

Variable	Real Information (RI)	False Information (FI)	t	*p*
Body Mass (kg)	77.1 ± 4.9	77.1 ± 5.0	−0.389	0.706
USG (a.u)	1.017 ± 0.007	1.017 ± 0.007	0.135	0.895
Uosm (mmol·kg^−1^)	654.3 ± 296.4	663.2 ± 297.4	0.279	0.786
Thirst perception (mm)	12.8 ± 10.8	14.1 ± 7.5	−1.38	0.199
WBGT (°C)	28.8 ± 0.1	28.9 ± 0.3	−0.814	0.461
Fullness	2.9 ± 1.0	2.9 ± 0.5	−1.27	0.232
Heat sensation	3.8 ± 1.0	3.7 ± 1.0	1.02	0.860

## Data Availability

Raw data for this research is available in the institutional repository: http://hdl.handle.net/10669/83765 (accessed 18 August 2021).

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
