# Peer review of "Awareness of Fluid Losses Does Not Impact Thirst during Exercise in the Heat: A Double-Blind, Cross-Over Study"

_nutrients, 2021, doi:10.3390/nu13124357_

Round 1

Reviewer 1 Report

Title / Line 45: Authors indicate this was a double-blind study. Authors should explain how double-blinding particularly for the investigators was accomplished given RI or FI feedback was given throughout the two exercise-heat trials. In other words, someone had to record changes in body weight and provide feedback (Lines 82-85) and thus were not blinded.  

Line 62: Authors should note the type of cloth’s worn during exercise-heat trials. Did they wear the same type of cloth’s during both trials?         

Line 68: The authors indicate the scale used - Accura model DSB291 – can measure to the nearest 10 grams. Is this correct? A scale measuring to this low of a weight suggests a table-top scale lab scale. I was unable to find any information about the scale on the internet. Authors should clarify that the model is indeed correct.

Line 81: Report room temperature and relative humidity in addition to WBGT.

Line 84: Authors should be specific on what “information” volunteers received. Was this information just fluid loss?  

Line 85: The Authors should comment on whether the fact that volunteers were told they would not be receiving water would influence TP responses? In other words, as noted in the introduction with regards to awareness of being observed, if the volunteers are aware they would not be able to drink water in either trials, would this information influence their TP response explaining why there was no difference in TP?  

Line 89: Authors should provide rationale for why during the post exercise-heat, water consumption was collected up to only 30 minutes? Could total water consumption be different had measures been made after 45 or 60 minutes? I may have missed it, but did the authors report water consumption between the groups at minute 15? Perhaps the total volume was not different at 30 minutes but the rate at which water was consumed was different? In other words, one group consumed more water than the other by minute 15. This would be interesting. 

On a similar note, water consumed post exercise-heat trials is underdeveloped in the discussion, only mentioned at the beginning (Line 139-141). Authors should expand the discussion in this area.  

Line 96: Was the VAS data (TP, fullness, and heat sensation) checked for normal distribution prior to performing an ANOVA?

Lines 93 and 102-104: Was an a priori power analysis performed or only a post hoc analysis? Authors should justify why they performed a post-hoc power analysis particularly since researchers have been encouraged not to perform such an analysis (https://doi.org/10.1002/pst.334). On a similar note, an effect size of 0.083, is this a Cohen’s or partial eta-squared effect size? From what data (i.e., previous study?) was this effect size derived? These information should be included.

Line 145-147: This could very well be the case, but because individuals typically will have access to fluids, not sure how this is relevant to your study where individuals were restricted from fluid intake during exercise.

Line 148-150: Figure 2A illustrates TP over time. Authors should determine the relationship between TP and dehydration to support their statements.  

Line 155-157: I am not clear as to the practical application of these data if individuals are typically not restricted from consuming fluids during exercise or competition.  

Line 155-157: Is the explanation simply that fluid was not consumed during exercise hence there was no difference in TP between the two groups? The explanation as to why there was no difference is underdeveloped – authors should expand on this.

Line 159-160: How practical is such statement? Are authors suggesting individuals not drink during exercise as means to monitor hydration status?

Line 160-162: Authors should be more specific on what they mean by “internal signals” and “inaccurate external information”. Provide references to support such statements.         

Title: Though I understand the Authors choice to use “water losses” in the title and throughout the manuscript, it is more accurate to say “sweat losses” or “fluid losses”. Authors should revise this.  

Volunteers appeared to complete both trials, RI and FI; the term cross-over should be included where relevant such as the in the title.

Reviewer 2 Report

The authors offer a novel study regarding the perception of thirst and awareness of fluid loss. The authors are commended for this work. 

Specific comments: 

Abstract, line 13: Is "heat stress" adequate for the measures taken? 

Intro, line 20: The authors should check "...a factor to take into consideration..." It is a factor for health and performance, right?

Procedures, line 59: "reported in other studies" but no citation offered? 

Procedures, lines 58,59: Why was "RI" and "FI" chosen to abbreviate trials? Not sure where the "I" comes from.

Procedures, 3rd paragraph: No citations offered for perceptual scales. The authors should consider backing up their methods. Have these been used in this form previously? Validity? 

Procedures, line 79: The reader here is left without details on 'baseline measurements.' What was taken, how, etc. needs to be added here. 

Procedures, line 82: Why was "near 4%" chosen for dehydration level? Seems like there should have been a goal at a fixed amount. 

Procedures: How, specifically, was the information on 'actual' losses shared? There are no details provided. 

Results, line 87: So, participants did not reach 4% dehydration. Why not?

Fig 1 vs. Fig 2: Trials are labeled as "Real" and "FALSE" in Fig 1, but "RWL" and "60WL" in Fig 2. These should be clear and consistent.

Discussion, line 147: Please clarify "..., while not caring much about performance." Athletes always care about performance?

Discussion, line 160: A discussion here about thirst being used as a parameter when water is withheld during exercise in the heat. There should be 1-2 citations here to support this statement. 

Discussion, line 161: This contradicts the results. "...internal signals seem to be adequate to indicate dehydration...", but, the authors found that thirst was mitigated soon after fluids were provided. Is thirst adequate for replacement?

Conclusion: Perhaps adding a statement regarding the adequacy of thirst or prescribed hydration to maximize health and sport performance in the heat could be added? It seems there may be evidence for prescribed hydration based on these results. This is an important clinical application?

Round 2

Reviewer 1 Report

Response 8: The authors consider that further discussion of this variable (post-exercise water intake) could distract the reader from the objective of the study: to identify if awareness of fluid loss could impact thirst perception during exercise in the heat.

Reviewer Comment: I understand the Authors point, though the same can be said for including the data. In other words, the readers may be interested in such data. Additionally, authors include make mention of such data in the last sentence in the first paragraph of the discussion which suggests some level of importance to the manuscript. The Authors should remove all aspects of the rehydration phase in the methods and results (Figures 1 and 2 A-D).  

Response 10: We can see a good argument in the Consultant’s Forum of the journal Pharmaceutical Statistics for why the power analysis would be better performed a-priori (for clinical trials). We must accept we were not aware of this paper; we relied instead on our experience with similar thirst studies which have found significant differences in thirst perception with a similar sample size). The effect size calculation is now described, lines 140-141.

Reviewer Comment: If I understand the Authors comments correctly, sample size for this study was determined from previous studies and not an a priori power analysis. If this is the case, the Authors should clearly state this in the manuscript. Including a post hoc power analysis now is out of context and provides no additional details.   

Response 12: We added % of dehydration to figure 2 (2B), to illustrated how TP and dehydration have the same behavior, line 190

Reviewer Comment: Adding the % dehydration figure still does not address my original comment on the relationship between dehydration and TP. In your data, what does the relationship look like?  

Reviewer 2 Report

The authors are commended for their corrections and additions. 

Author Response

No comments from reviewer 2